# CMOS Voltage Reference Using a Self-Cascode Composite Transistor and a Schottky Diode

**Thaironi M. Brito [1],\*, Dalton M. Colombo [2] 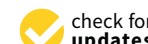, Robson L. Moreno [1] and Kamal El-Sankary [3],\***

[1] Systems Engineering and Information Technology Institute, Federal University of Itajuba, 35903-087 Itajuba, Brazil; moreno@unifei.edu.br

[2] Electrical Engineering Department, Federal University of Minas Gerais, 31270-901 Belo Horizonte, Brazil; daltonmc@ufmg.br

[3] Electrical & Computer Engineering Department, Dalhousie University, Halifax, NS B3J 1Z1, Canada

\* Correspondence: thaironi.menezes@unifei.edu.br (T.M.B.); Kamal.El-Sankary@dal.ca (K.E.-S.)

**Abstract:** This work presents an investigation of the temperature behavior of self-cascode composite transistors (SCCTs). Results supported by silicon measurements show that SCCTs can be used to generate a proportional to absolute temperature voltage or even a temperature-compensated voltage. Based on the achieved results, a new circuit topology of a resistorless voltage reference circuit using a Schottky diode is also presented. The circuit was fabricated in a 130 nm BiCMOS process and occupied a silicon area of 67.98 μm × 161.7 μm. The averaged value of the output voltage is 720.4 mV, and its averaged line regulation performance is 2.3 mV/V, calculated through 26 characterized chip samples. The averaged temperature coefficient (TC) obtained through five chip samples is 56 ppm/°C in a temperature range from −40 to 85 °C. A trimming circuit is also included in the circuit topology to mitigate the impact of the fabrication process effects on its TC. The circuit operates with a supply voltage range from 1.1 to 2.5 V.

**Keywords:** voltage reference; Schottky diode; self-cascode composite transistor; resistorless

## 1. Introduction

The voltage reference circuit, as is well known, provides an output voltage ($V_{REF}$) that is insensitive to variations in the supply voltage ($V_{DD}$), operation temperature, and the fabrication process effects. The temperature-compensated $V_{REF}$ is often achieved by the balanced addition of two voltages with opposite temperature coefficients: VPTAT (proportional to absolute temperature) and VCTAT (complementary proportional to absolute temperature). The CTAT voltage is usually implemented by means of the base-emitter voltage ($V_{BE}$) or the gate-source voltage ($V_{GS}$). The PTAT voltage is typically generated by the difference between two $V_{BE}$ or $V_{GS}$ voltages.

A non-conventional approach to generate the PTAT voltage for a voltage reference application was proposed by [1]. It uses two transistors in series with both gate terminals tied together—an association here called self-cascode composite transistors (SCCTs). This approach has shown promise and has been employed in other works, for instance, in [2–4]. Therefore, in order to extract the best use of this type of transistor association, it is essential to investigate its temperature dependence.

A non-conventional approach to generate the CTAT voltage was proposed in [5]. The use of a Schottky diode to generate the CTAT voltage allows for the reduction of the minimum supply voltage of circuit operation. The Schottky diode was also successfully employed in [3,6]. The use of $V_{GS}$ voltage to implement the CTAT voltage also allows for VDD reduction. However, the generated $V_{REF}$ in this type of topology is directly proportional to the transistor threshold voltage, $V_{TH}$, which is a

strong function of process parameters. The doping at the transistor channel region, for example, is hard to control in the fabrication of downscaled technology.

This work studies the temperature dependency of SCCTs in triode and saturation mode. It is shown that the output voltage of this device can have PTAT or temperature-compensated behavior, as supported by silicon measurements. Based on these results, a new topology of CMOS resistorless voltage reference using a Schottky diode [3] was designed and validated with silicon measurements. Additionally, a trimming circuit was included in the circuit topology to mitigate the impact of fabrication process effects on its temperature performance.

The proposed circuit generates an output voltage with an average value of 720.4 mV and a standard deviation (σ) of 15.6 mV, calculated considering 26 measured samples. The averaged TC after trimming is 56 ppm/°C considering five chip samples. Moreover, the circuit operates with a $V_{DD}$ range from 1.1 to 2.5 V and occupies a silicon area of 67.98 μm × 161.7 μm.

This paper is organized as follow. Section 2 describes the generation of the CTAT and PTAT voltages. The proposed circuit topology is discussed in Section 3. Sections 4 and 5 present the silicon results and conclusions, respectively.

## 2. CTAT and PTAT Voltage Generators

### 2.1. The Schottky Diode as a $V_{CTAT}$ Generator

Unlike the PN-junction diode, the Schottky diode has a metal-semiconductor junction. This junction is obtained taking metal into contact with a moderately doped n-type diffusion. As a result, the Schottky diode differs from the PN diode in two electrical characteristics: fast switching and a low forward-voltage drop ($V_D$), which is about 300 mV. The second characteristic is important for the voltage reference design operation at a low supply voltage. The Schottky diode current equation is given below [5]:

$$I_D = I_S\left(\exp\left\{\frac{V_D}{nU_T}\right\} - 1\right) \tag{1}$$

where $q$ is the electron charge, $n$ is the slope factor, and $U_T$ is the thermal voltage.

The Schottky diode current equation is similar to the PN-junction current equation, which indicates that it may be possible to replace the PN diode by the Schottky diode. However, it is vital to analyze Schottky temperature behavior before using it in place of a PN diode. Figure 1 shows a comparison between the simulated forward biasing voltages, for the same current density level, of a Schottky diode and a vertical bipolar transistor. The simulated TC of the Schottky diode and PN diode is 1.46 and 1.92 mV/°C, respectively. Therefore, it is possible to use the Schottky diode to generate the CTAT voltage.

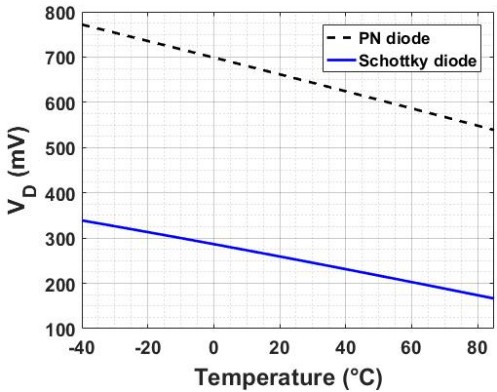

**Figure 1.** Simulated $V_D$ vs. temperature for the Schottky diode and the PN-junction diode.

### 2.2. The Self-Cascode Composite Transistor (SCCT) As a $V_{PTAT}$ Generator

The SCCT is the association of two transistors, as can be seen in Figure 2 [2]. Both substrate terminals are connected to the ground reference.

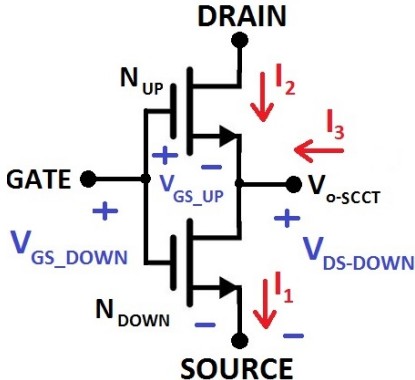

**Figure 2.** The self-cascode composite transistor (SCCT).

The drain current of a transistor operating in sub-threshold weak inversion is given by Equation (2) [7]:

$$I_D = I_0 \exp\{\frac{V_{GS} - Vt}{nU_T}\}(1 - \exp\{-\frac{V_{DS}}{U_T}\})$$ (2)

where $I_0$ is given by [7]

$$I_0 = \mu_0 C_{ox} \frac{W}{L}(n-1)Ut^2$$ (3)

where $\mu_0$ is the carrier mobility, and $C_{OX}$ is the oxide capacitance. From Equations (2) and (3), the gate-source voltage ($V_{GS}$) is written as

$$V_{GS} = nU_T \ln\left(\frac{I_D}{u_o C_{OX} S(n-1)U_T^2 * [1 - \exp\{-\frac{V_{DS}}{U_T}\}]}\right) + V_t$$ (4)

where $S$ is the transistor aspect ratio (W/L). The output voltage of the SCCT, $V_{DS-DOWN}$, is the difference between the gate-source voltages of both transistors:

$$V_{DS-DOWN} = V_{GS-DOWN} - V_{GS-UP}.$$ (5)

Replacing Equation (4) in Equation (5), the drain-source voltage of transistor $N_{DOWN}$ is

$$V_{DS_{down}} \approx nU_T \ln\left(\frac{I_{D_{down}} S_{up}[1 - \exp\{-\frac{V_{DSup}}{U_T}\}]}{I_{D_{up}} S_{down}[1 - \exp\{-\frac{V_{DS_{down}}}{U_T}\}]}\right).$$ (6)

When the drain-source voltage is more than about four times the thermal voltage ($V_{DS} \geq 4U_T$), Equation (6) can be approximated by

$$V_{DS_{down}} \approx nU_T \ln\left(\frac{I_{D_{down}} S_{up}}{I_{D_{up}} S_{down}}\right).$$ (7)

Equations (6) and (7) consider an equal threshold voltages of $N_{UP}$ and $N_{DOWN}$, and that neglects the small error caused by the body effect in the $N_{UP}$ transistor.

To characterize the temperature behavior of the SCCTs, and its dependency on the current level, the circuit in Figure 3 was designed and fabricated. This circuit includes several SCCTs composed by transistors with different aspect ratios. Each SCCT device is a combination in series or in parallel

of a unitary transistor with an aspect ratio of 3 μm/3 μm. The unitary transistor was designed with a length equal to 3 μm for a 130 nm process with the intention of (i) avoiding short-channel effects and (ii) mitigating the impact of fabrication process effects on the output voltage. The aspect ratios of transistors P1–P9 in Figure 3 are the same to provide the same biasing current in the SCCTs. With reference to Figure 2, $I_1 = I_2$, and $I_3 = 0$ for each device.

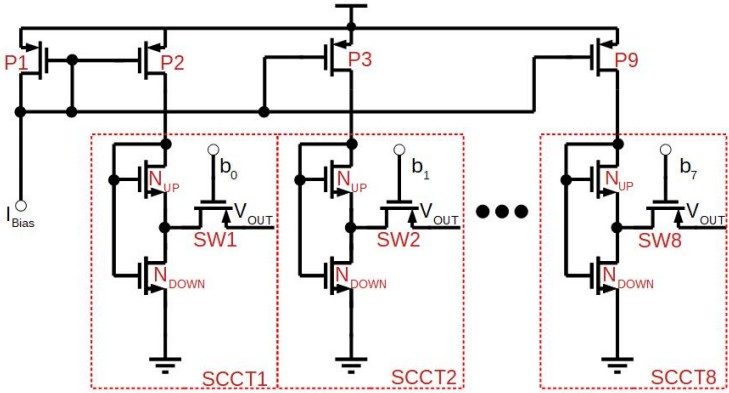

**Figure 3.** The matrix of SCCTs.

Eight different combinations of SCCTs were designed, with different W/L aspect ratios. The objective was to investigate and validate, through silicon measurements, the temperature behavior predicted by simulations. For instance, transistor $N_{UP}$ in SCCT1 and SCCT8 have an equivalent W/L equal to 60 μm/3 μm, and 42 μm/3 μm, respectively. These transistor dimensions were chosen to set the inflection point of the output voltage vs. the temperature curve at a specific temperature value. For instance, the inflection points for SCCT1 and SCC8 are about 18 and 0 °C, respectively.

Each SCCT was measured in a temperature range from −40 to 85° for three temperature-independent bias currents: 100 nA, 1 μA, and 100 μA. The supply voltage was 1.2 V, and the measurement results are presented in the next section.

## 2.3. Silicon Measurements of the Output Voltage of SCCTs

Figure 4 shows the output voltage of SCCT1 and SCCT8 as a function of temperature for a bias current of 100 nA. The output voltage of the SCCT1 is larger than the SCCT8 because of its larger ratio of $S_{up}/S_{down}$ as expected by Equation (6). Moreover, the inflexion point of both curves is different, as mentioned in the last section. This result shows that the designer can use the proper transistor sizing in order to adjust the value of the output voltage and its temperature coefficient.

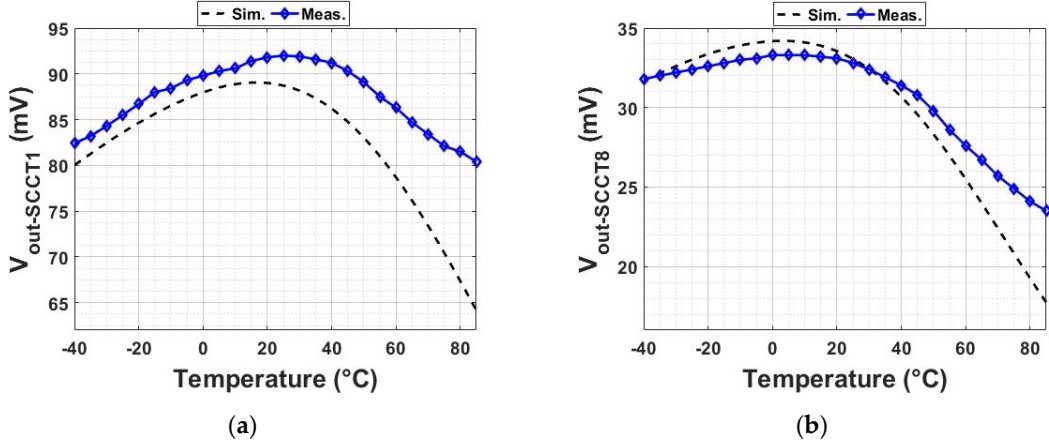

**Figure 4.** Measured output voltage as a function of temperature for the (**a**) SCCT1 and (**b**) SCCT8 using $I_{Bias}$ of 100 nA.

Figure 5 shows the output voltage of SCCT1 and SCCT8 as a function of temperature for a bias current of 1 µA. As shown in Figures 4 and 5, the higher the bias current, the greater the positive slope (TC) of the output voltage is. This means that the curve is no longer a downward concave; it tends to be a straight line. This result is also expected from Equation (6).

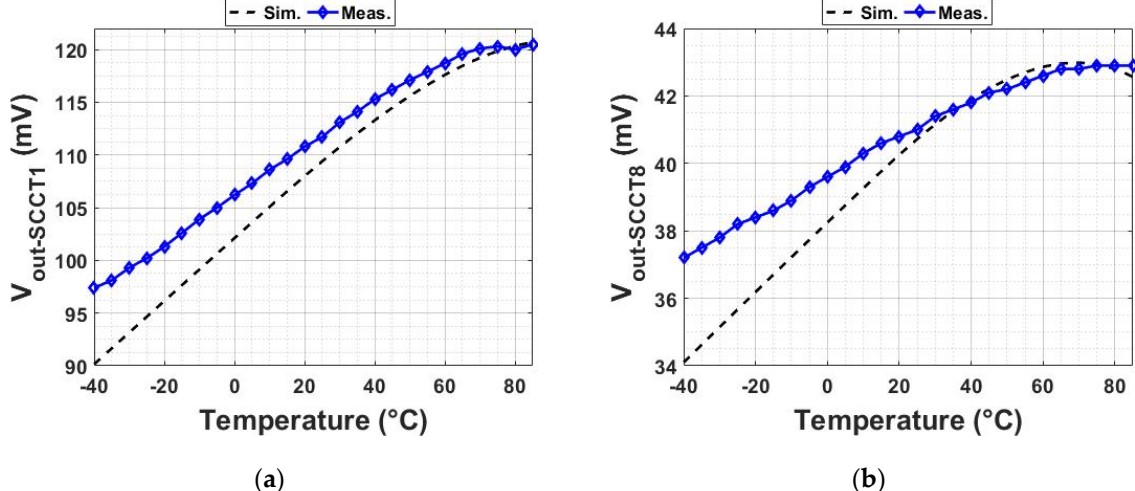

**Figure 5.** Measured output voltage as a function of temperature for (**a**) SCCT1 and (**b**) SCCT8 using $I_{Bias}$ of 1 µA.

Figure 6 shows the output voltage of SCCT1 and SCCT8 as a function of temperature for a bias current of 10 µA. Figures 4–6 show that, as the bias current is increased, the temperature dependence of the output voltage tends to be a rising linear curve. This temperature relation was observed in all measured SCCTs. The other temperature results were compiled in Table 1, while the aspect ratio of all SCCTs is given in Table 2. Moreover, it was verified through simulations that the linear temperature dependence continues to occur for bias currents from 10 µA to 1 mA.

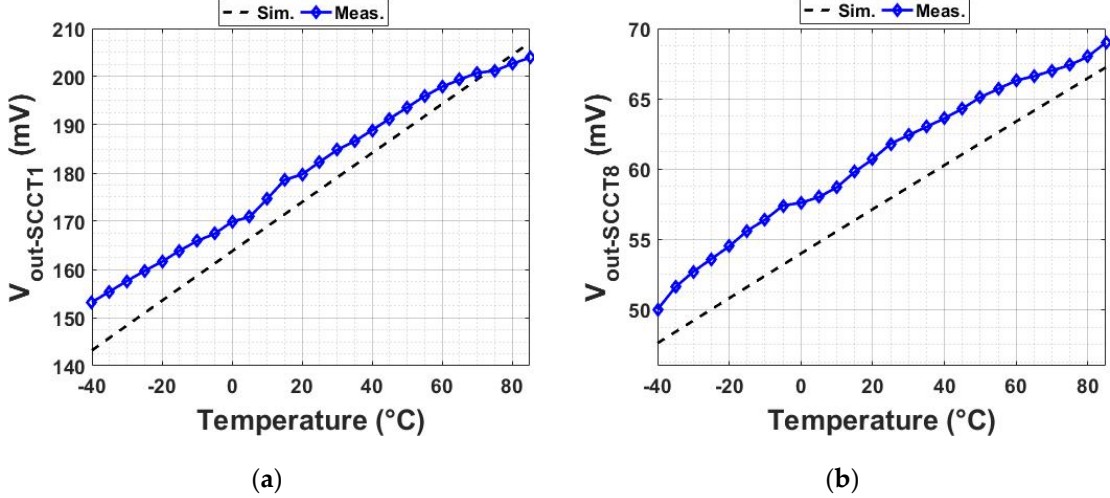

**Figure 6.** Measured output voltage as a function of temperature for (**a**) SCCT1 and (**b**) SCCT8 using $I_{Bias}$ of 10 µA.

**Table 1.** Measured temperature coefficients (TCs) of the SCCTs.

| | | TC (ppm/°C) | | | | | | | |
|---|---|---|---|---|---|---|---|---|---|
| **I_BIAS** | **Data** | **Self-Cascode Composite Transistors** | | | | | | | |
| | | SCCT1 | SCCT2 | SCCT3 | SCCT4 | SCCT5 | SCCT6 | SCCT7 | SCCT8 |
| 100 nA | Sim | 2200 | 1900 | 1400 | 1400 | 2300 | 1300 | 3300 | 4000 |
| | Meas | 1000 | 822 | 1600 | 662 | 1100 | 1100 | 1600 | 2400 |
| 1 µA | Sim | 2200 | 2300 | 2700 | 2500 | 2200 | 2600 | 1900 | 1700 |
| | Meas | 1600 | 1600 | 2000 | 1500 | 1500 | 1700 | 1700 | 1100 |
| 10 µA | Sim | 2900 | 2900 | 2800 | 2800 | 2800 | 2800 | 2700 | 2700 |
| | Meas | 2200 | 2400 | 2100 | 2200 | 2100 | 2300 | 2100 | 2500 |

**Table 2.** Transistor aspect ratios and operation voltages of all SCCTs at T = 27 °C.

| Self-Cascode | Transistor | Aspect Ratio | $V_{GS}$ (mV) | | | $V_{DS}$ (mV) | | | Mode | | |
|---|---|---|---|---|---|---|---|---|---|---|---|
| | | | 100 nA | 1 µA | 10 µA | 100 nA | 1 µA | 10 µA | 100 nA | 1 µA | 10 µA |
| SCCT1 | $N_{UP}$ | $(20 \cdot 3\mu)/3\mu$ | 42.5 | 117.5 | 210.1 | 42.5 | 117.5 | 210.1 | Trio. | Sat. | Sat. |
| | $N_{DOWN}$ | $3\mu/3\mu$ | 131.1 | 227.5 | 387.6 | 88.5 | 109.9 | 117.5 | Trio. | Sat. | Sat. |
| SCCT2 | $N_{UP}$ | $(14 \cdot 3\mu)/3\mu$ | 51.3 | 129 | 223.8 | 51.3 | 129 | 223.8 | Trio. | Sat. | Sat. |
| | $N_{DOWN}$ | $3\mu/3\mu$ | 132 | 228.6 | 389.6 | 80.6 | 99.6 | 165.8 | Trio. | Trio. | Sat. |
| SCCT3 | $N_{UP}$ | $(4 \cdot 3\mu)/3\mu$ | 83.3 | 175.4 | 285.1 | 89.36 | 175.4 | 285.1 | Trio. | Sat. | Sat. |
| | $N_{DOWN}$ | $3\mu/(2 \cdot 3\mu)$ | 140.8 | 246.6 | 405.9 | 51.44 | 65.19 | 120.7 | Trio. | Trio. | Sat. |
| SCCT4 | $N_{UP}$ | $(8 \cdot 3\mu)/3\mu$ | 66.4 | 147.5 | 247.3 | 66.4 | 147.5 | 247.3 | Trio. | Sat. | Sat. |
| | $N_{DOWN}$ | $3\mu/3\mu$ | 134 | 231.2 | 394.3 | 67.5 | 83.7 | 147 | Trio. | Trio. | Sat. |
| SCCT5 | $N_{UP}$ | $(10 \cdot 3\mu)/3\mu$ | 57.9 | 136.7 | 233.1 | 57.9 | 136.7 | 233.1 | Trio. | Sat. | Sat. |
| | $N_{DOWN}$ | $(2 \cdot 3\mu)/3\mu$ | 111 | 201.2 | 337 | 53 | 64.4 | 103.9 | Trio. | Trio. | Sat. |
| SCCT6 | $N_{UP}$ | $(4 \cdot 3\mu)/3\mu$ | 88.7 | 171.6 | 281.2 | 88.7 | 171.6 | 281.2 | Trio. | Sat. | Sat. |
| | $N_{DOWN}$ | $3\mu/3\mu$ | 138 | 236.7 | 404.5 | 51.2 | 65.1 | 123.2 | Trio. | Trio. | Sat. |
| SCCT7 | $N_{UP}$ | $(12 \cdot 3\mu)/3\mu$ | 51.1 | 127.9 | 221.5 | 51.1 | 127.9 | 221.5 | Trio. | Sat. | Sat. |
| | $N_{DOWN}$ | $(4 \cdot 3\mu)/3\mu$ | 90.3 | 175.9 | 293.2 | 39.2 | 48 | 71.6 | Trio. | Trio. | Trio. |
| SCCT8 | $N_{UP}$ | $(14 \cdot 3\mu)/3\mu$ | 46.1 | 121.3 | 213.2 | 46.1 | 121.3 | 213.2 | Trio. | Sat. | Sat. |
| | $N_{DOWN}$ | $(6 \cdot 3\mu)/3\mu$ | 79 | 162.2 | 271.4 | 32.8 | 41 | 58.2 | Trio. | Trio. | Trio. |

Table 1 presents the temperature coefficient given in ppm/°C calculated using Equation (8). Sim and Meas stand for simulation and measurement, respectively. Parameter $V_{out}$ is measured at 27 °C, $V_{out\_max}$ and $V_{out\_min}$ represent the maximum and minimum values of the output voltage in the temperature range, respectively. For instance, the measured data of Figure 4a of $V_{out}$ = 91.5 mV, $V_{out\_max}$ = 92 mV, $V_{out\_min}$ = 80 mV, $T_{max}$ = 85 °C, and $T_{min}$ = −40 °C yield a TC of about 1000 ppm/°C.

$$TC = \frac{1}{V_{out}} \frac{\left(V_{out\_max} - V_{out\min}\right)}{T_{max} - T_{min}} \cdot 10^6 \left[\frac{ppm}{°C}\right]. \tag{8}$$

For currents around 100 nA (Figure 4), the output voltage tends to be temperature-compensated and can be used a standalone voltage reference without the need of an auxiliary CTAT circuit. While in this case, the output voltage presents a more significant temperature curvature when compared to output generated by traditional voltage references, this is a very important result owing to the simplicity of the SCCT circuit. Hence, SCCTs can be used to create an output voltage with some temperature compensation, for moderate accuracy applications, while using a minimal silicon area and a low power consumption.

As shown in Figure 6, the output of SCCTs presents a linear PTAT TC when biased with a current greater than 10 µA. It is also possible to generate a rising linear output voltage for currents lower than 10 µA by increasing the ratio of $S_{UP}/S_{DOWN}$ in Equation (7) at the expense of a larger silicon area.

Table 2 presents the operation mode of each transistor of the SCCT where Trio and Sat stand for triode and saturation operation, respectively. As can be seen, by increasing the bias current, the transistor leaves the linear mode and enters saturation mode. Note that, for $I_{BIAS}$ equal to 100 nA, all transistors are in weak inversion operation, i.e., the gate-source voltage is lower than the threshold voltage. The threshold voltage for all transistors in Table 2 is about 210 mV. The condition to be in

saturation with weak inversion biasing is the drain-source voltage larger than about four times the thermal voltage (~100 mV @ 22 °C).

Consider, for instance, Figure 5. The output voltage of both SCCT1 and SCCT8 presents a linear PTAT dependency up to 60 °C and a lower slope after that. This happens because, as the output voltage increases with temperature, the drain-source voltage of the upper transistor decreases and eventually enters a triode region at 85 °C. For $I_{BIAS}$ equal to 10 µA, the drain-source voltage is high enough to keep the upper transistor of all SCCTs in saturation. Therefore, if the linear PTAT TC is desired, the upper transistor must be in saturation mode, while the lower transistor can be in triode or saturation mode.

If the temperature-compensated output voltage is desired, the designer can properly bias the SCCT with low currents. However, it is important to be aware that the output voltage of the SCCT and its derivative are strongly dependent on the bias current for values lower than 5 µA, as shown in Figure 7 for SCCT1.

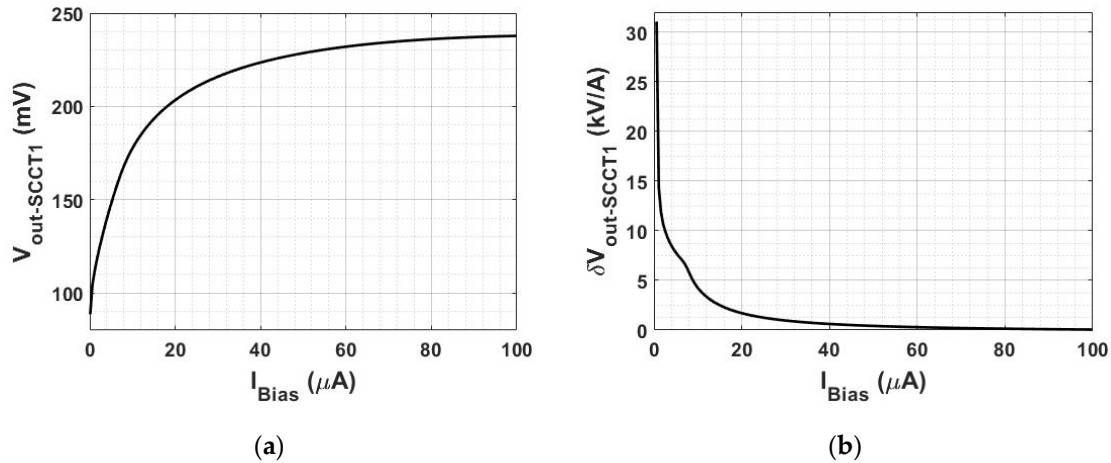

(**a**)　　　　　　　　　　　　　　　　　　　　　　　　(**b**)

**Figure 7.** (**a**) $V_{out}$ of SCCT1 and its (**b**) derivative as function of $I_{Bias}$.

Figure 8 shows the TC of the SCCT output voltage as a function of the bias current. The TC is a strong function of the bias current when the upper transistor is in triode mode, i.e., an $I_{BIAS}$ lower than 10 µA. This strong dependence imposes a challenge in using the temperature-compensated output voltage. The reason for that is the difficulty of generating an accurate and stable bias current insensitive to fabrication process effects. Moreover, the TC of the bias current also modifies the TC of the output voltage. Therefore, if the chip already has an accurate current reference circuit, a low area reference voltage with some temperature compensation can be efficiently designed using SCCTs. Finally, the SCCT operating in triode mode acts as a moderate temperature-compensated resistor.

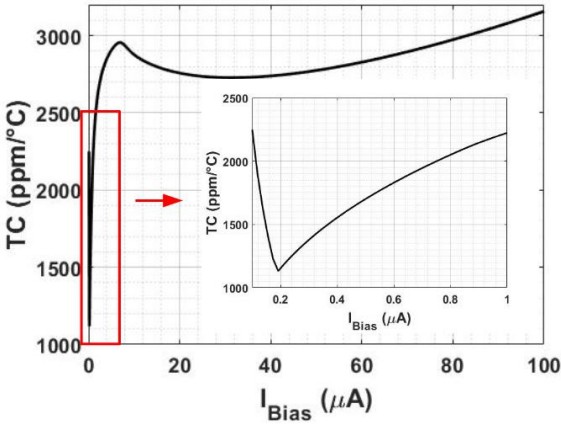

**Figure 8.** The TC of SCCT1 as a function of $I_{BIAS}$.

### *2.4. Comparison to Other Works*

Several works have used SCCT devices. Some works [4,8–11] show that the SCCTs can also generate a moderate temperature-compensated output voltage. Some work [4] explores the temperature compensation of SCCT utilizing two transistors with different gate oxide thicknesses and different threshold voltages. The $N_{UP}$ is a thick gate oxide transistor, while the $N_{DOWN}$ is a thin-gate oxide transistor, respectively. A similar approach is also used in [8,9]. In Reference [10], an alternative configuration of the SCCT is proposed, where the gate terminal of SCCT is connected to the output voltage. Moreover, the bulk terminal of $N_{UP}$ is also connected to the output voltage to eliminate the body effect. That solution requires a CMOS triple-well technology.

In Reference [11], $N_{UP}$ and $N_{DOWN}$ devices have the same gate-oxide thickness but with different lengths. Transistors with different lengths have different threshold voltages, mainly caused by the reverse short-channel effect (RSCE).

The study and the experimental data presented in this work provide general guidelines to the use of SCCTs in the design of voltage references. In temperature-compensated reference $V_{REF}$ using CTAT and PTAT circuits, the SCCTs can be used as the PTAT block when $V_{OUT\_SCCT}$ has a linear dependence to temperature. This work also shows that a moderate temperature-compensated output voltage can be generated using an SCCT with only two transistors of the same gate oxide thicknesses and the same transistor length operating in triode mode.

## 3. The Proposed Voltage Reference

The proposed circuit is shown in Figure 9. It is composed of two sub-circuits: the bias current generator and the voltage reference core. The substrate terminal of all NMOSFETs and the n-well terminal of all PMOSFETs are connected to ground and supply, respectively. The bias current generator was proposed in [12] and is formed by transistors P1, P2, P3, N1, N2, N3, N4, and the amplifier A1. Transistor N4 operates in the triode region and acts as an integrated resistor. Its drain-source voltage ($V_{DSN4}$) is equal to $V_{GSN2} - V_{GSN1}$, and the current ($I_1$) flowing through its terminals can be approximated by Equation (9). $I_1$ is also given by Equation (10) [12]:

$$I_1 = \mu_0 C_{OX}\left(\frac{W}{L}\right)(V_{GSN4} - V_{TH})V_{DSN4} \tag{9}$$

$$I_1 = n_c^2 \beta_{N4} U_T^2 K_{eff} \tag{10}$$

$$K_{eff} = \left[K_2 - 0.5 + \sqrt{K_2(K_2 - 1)}\right]\ln^2(K_1) \tag{11}$$

$$K_1 = \frac{S_{N1}S_{P2}}{S_{N2}S_{P1}} \tag{12}$$

$$K_2 = \frac{S_{N4}S_{P3}}{S_{N3}S_{P1}} \tag{13}$$

where $\beta$ is the transconductance parameter, and $n_C$ is a correction factor for low drain-source voltages [12]. Note that I1 depends mainly on the aspect ratio of transistors N1–N4, P1–P3, mobility, and $U_T$. Moreover, the performance temperature of $I_1$ is given by [12]

$$I_1 = \left(n_c^2 \beta_{N4}\right)_0 U_{T0}^2 K_{eff}\left(\frac{T}{T_0}\right)^{2-m} \tag{14}$$

where subscript 0 means room temperature, and parameter m is 1.5 and 2 [12]. The bias current in our circuit was designed to be 15 nA. Transistor P8 was included to allow the temperature characterization of $I_{REF}$. The temperature coefficient of $V_{REF}$ depends on the TC of $I_{REF}$. Transistor P8 was designed with a large width to increase the value of the test current.

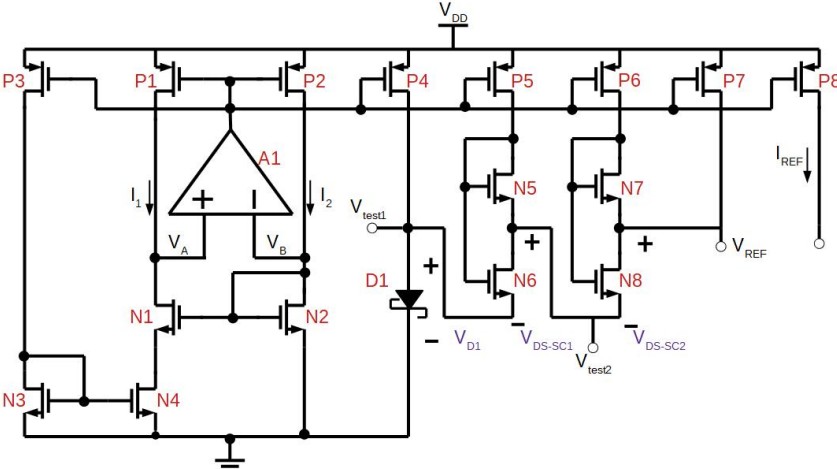

**Figure 9.** The proposed voltage reference.

Referring to Figure 9, D1 is a Schottky diode, and it is responsible for generating the CTAT voltage. The PTAT voltage is generated through two SCCTs. The first one is composed of transistors N5–N6, while the second one is composed of transistors N7–N8.

The output voltage of the designed circuit is given by

$$V_{REF} = V_{D1} + V_{DSN5} + V_{DSN7}. \tag{15}$$

Since both SCCTs operate in weak inversion and saturation, it is possible to rewrite the drain-source voltages using Equation (7), and the output voltage can be described by Equation (16):

$$V_{REF} = V_{D1} + nU_t \ln\left[\frac{S_{N5}S_{N7}(S_{P5} + S_{P6} + S_{P7})(S_{P6} + S_{P7})}{S_{N6}S_{N8}S_{P5}S_{P6}}\right]. \tag{16}$$

According to Equation (16), the desired temperature compensation is achieved with the proper sizing of the current mirror and the SCCTs. Table 3 shows the dimensions of all transistors. Except for the SCCTs (N5-N8), which are thin-oxide 1.2 V transistors, all others are 2.5 V thick oxide transistors. The use of thick-oxide transistors is for a better line regulation. It would also be possible to use thick-oxide transistors for both SCCTs, but it would result in an extra layout area. Finally, the pins $V_{test1}$ and $V_{test2}$ were included to characterize the designed circuit.

**Table 3.** Transistor dimensions of the proposed voltage reference.

| Transistor | P1 | P2 | P3 | P4 | P5 | P6 | P7 | P8 | P9 | P10 |
|---|---|---|---|---|---|---|---|---|---|---|
| W(μm) | 1 | 1 | 1 | 1 | 1 | 1 | 1 | 1 | 0.4 | 0.4 |
| L(μm) | 14 | 14 | 14 | 14 | 14 | 14 | 14 | 14 | 20 | 20 |
| Parallel | 2 | 2 | 4 | 2 | 40 | 16 | 2 | 14 | 2 | 2 |
| Series | 1 | 1 | 1 | 1 | 1 | 1 | 1 | 1 | 1 | 1 |
| **Transistor** | **N1** | **N2** | **N3** | **N4** | **N5** | **N6** | **N7** | **N8** | **N9** | **N10** |
| W(μm) | 2 | 2 | 0.6 | 0.6 | 1.4 | 1.4 | 3.9 | 3.9 | 0.4 | 0.4 |
| L(μm) | 9 | 9 | 20 | 20 | 1 | 1 | 1 | 1 | 20 | 20 |
| Parallel | 8 | 1 | 1 | 1 | 18 | 1 | 4 | 1 | 2 | 2 |
| Series | 1 | 1 | 3 | 3 | 1 | 24 | 1 | 22 | 1 | 1 |

### 3.1. Amplifier Circuit

The schematic of the amplifier A1 in Figure 9 is shown in Figure 10. It is used to improve the line regulation of the current bias generator [13], and its nominal open-loop voltage gain at DC is 55 dB.

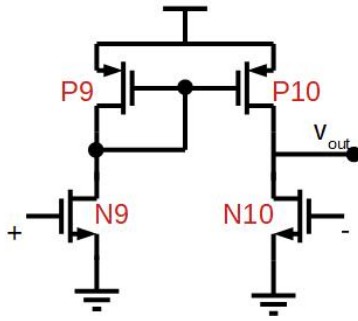

**Figure 10.** Amplifier A1.

### 3.2. Trimming Circuit

The impact of the fabrication process effects on circuit performance was predicted by Monte Carlo analysis considering 1000 samples. Transistor mismatch and process variations were considered in this simulation, and the histogram for $V_{REF}$ and the TC of $V_{REF}$ are shown in Figure 11.

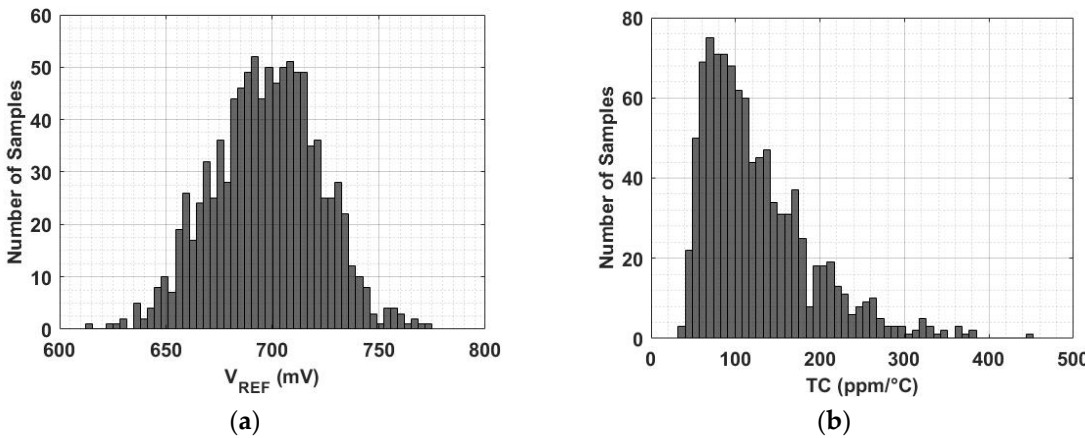

(**a**)                                                 (**b**)

**Figure 11.** Histogram of (**a**) $V_{REF}$ and (**b**) the TC of $V_{REF}$.

The mean value and the standard deviation of $V_{REF}$ at 27 °C are 697.1 mV and 25.11 mV, respectively. Regarding the temperature coefficient of $V_{REF}$, its mean value and σ are 125 and 63.78 ppm/°C, respectively. As discussed in Section 2, the output voltage produced by SCCTs is dependent on the bias current. Thus, it is also important to simulate the electrical variability of the bias current. The mean value and standard deviation of $I_{REF}$ at 27 °C is 103.4 and 14 nA, respectively.

Based on Figure 11b, the TC of $V_{REF}$ can be severely degraded by the fabrication process variability. To mitigate the performance degradation of the TC, the trimming circuit shown in Figure 12 was included in the reference circuitry. Table 4 presents all the transistor dimensions of the trimming circuit.

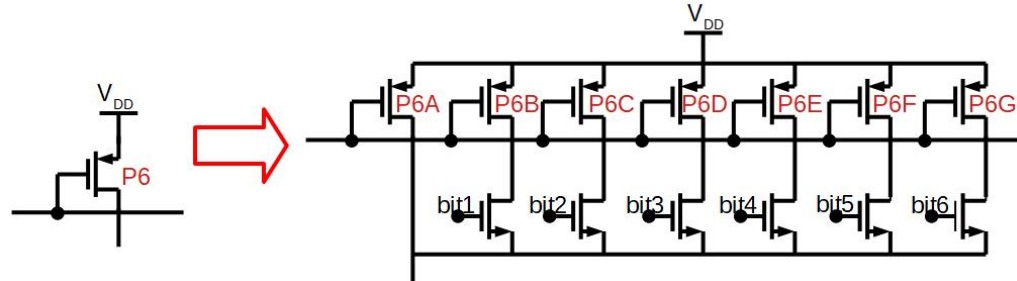

**Figure 12.** Trimming circuit.

**Table 4.** Transistor dimensions of the trimming circuit.

| Transistor | P6A | P6B | P6C | P6D | P6E | P6F | P6H |
|---|---|---|---|---|---|---|---|
| W(μm) | 1 | 1 | 1 | 1 | 1 | 1 | 1 |
| L(μm) | 14 | 14 | 14 | 14 | 14 | 14 | 14 |
| Parallel | 4 | 4 | 4 | 4 | 4 | 4 | 4 |
| Series | 1 | 1 | 1 | 1 | 1 | 1 | 1 |

According to the measured results of SCCTs (Section 2.2), the more current flowing through the SCCT, the greater the positive slope (TC) of the output voltage. Therefore, if $V_{REF}$ is not temperature-compensated, controlling the current level of SCCTs allows for the adjustment of the TC of $V_{REF}$.

The current flowing through the SCCT composed by N7 and N8 was chosen to be adjusted. The control of the current level was obtained, splitting the transistor P6 into seven equal transistors. Six of them have a switch that is controlled by a digital signal external to the chip. The proposed voltage reference was designed with three switches at a high level (on-state) and three at low-level (off-state). If it is needed to increase the PTAT component of $V_{REF}$, more switches must be turned on. If the opposite is needed, then fewer switches must be turned on.

Figure 13 presents the simulated $V_{REF}$ versus temperature as a function of the number of switches in the on-state. As can be seen, few switches are needed to cover a considerable variation of $V_{REF}$ and its TC. The implemented trimming circuit is overdesigned in this chip. Table 5 shows the value of $V_{REF}$ at 27 °C, and shows its TC as a function of the number of switches in on-state.

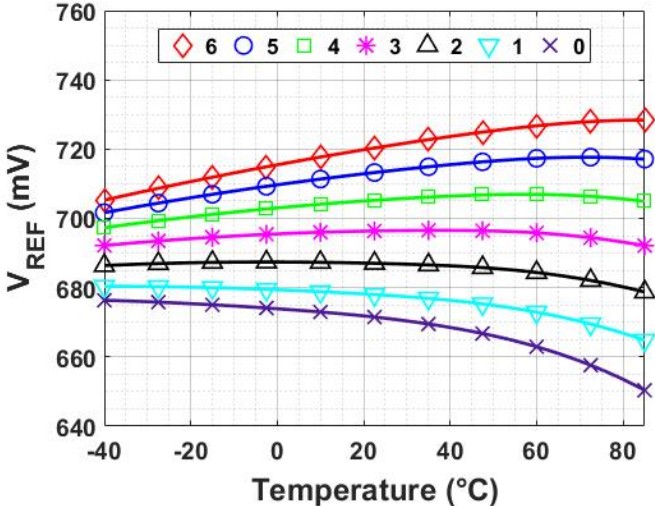

**Figure 13.** Simulated $V_{REF}$ as a function of temperature and the number of switches at a high level.

**Table 5.** $V_{REF}$ and TC as a function of the number of switches at a high level.

| Number of Switches | $V_{REF}$ (mV) @ 27°C | TC (ppm/°C) | Temperature Performance |
|---|---|---|---|
| 0 | 670.8 | 310.1 | CTAT |
| 1 | 677.6 | 184 | CTAT |
| 2 | 686.87 | 100.8 | CTAT |
| 3 | 696.4 | 50.8 | Compensated |
| 4 | 705.62 | 109.3 | PTAT |
| 5 | 713.9 | 179.4 | PTAT |
| 6 | 721.3 | 256.3 | PTAT |

### 3.3. Start-Up Circuit

The voltage reference includes a start-up circuit shown in Figure 14. It prevents the bias current generator stabilizes in a state with all currents being equal to zero. The start-up circuit works as follows.

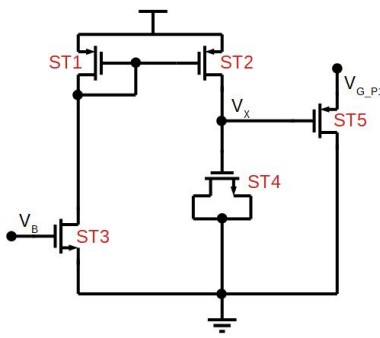

**Figure 14.** The start-up circuit.

When I1 and I2 (Figure 9) are equal to zero, the gate voltages of transistors P1 and P2 are close to $V_{DD}$, and the gate voltage of N1 and N2 are close to 0 V. Thus, the transistor ST3 are in the cut-off region, and there is no current being mirrored by ST2. The transistor ST4, which is operating as a capacitor, is therefore not charged. This means that node Vx remains equal to 0 V. At this moment, the source-gate voltage of ST5 is nearly VDD, and transistor ST5 starts to drain the current from the gate node of the P1 transistor (Figure 9).

As a current is drawn from this node, the gate voltage of P1 is decreased, and a current starts to flow through the P1 and P2 devices. When the bias generator is operating, ST3 is no longer turned off, and ST4 starts charging node $V_x$ until ST5 enters cut-off. At this moment, the start-up circuit no longer has any influence on the bias current generator. The dimensions of all transistors of the start-up circuit are shown in Table 6.

**Table 6.** Transistors dimensions of the start-up circuit.

| Transistor | ST1 | ST2 | ST3 | ST4 | ST5 |
|---|---|---|---|---|---|
| W(μm) | 1 | 1 | 0.36 | 5 | 5 |
| L(μm) | 5 | 5 | 20 | 5 | 0.50 |
| Parallel | 2 | 2 | 1 | 1 | 1 |
| Series | 1 | 1 | 2 | 1 | 1 |

## 4. Silicon Results

A microphotograph of the fabricated chip can be seen in Figure 15. The voltage reference circuit occupies a small area of 10,992 μm$^2$. The output voltage and current were measured under temperature and supply voltage variations. Figure 16 shows the temperature measurement of $V_{REF}$ in a range from −40 to 85 °C, considering five chip samples.

As can be seen, S4 and S5 samples work precisely as desired. However, S1, S2, and S3 do not have the maximum nominal temperature compensation. While S1 and S2 have PTAT behavior, S3 has a CTAT behavior. The Monte Carlo analysis predicted the deviation from the nominal performance.

Using the trimming circuit, it was possible to improve the temperature performance of S1, S2, and S3. In S1 and S2, two switches were turned off, which means that only one switch was maintained at a high level. In S3, one switch was turned on, resulting in four switches at a high level. The output voltage, as a function of temperature after circuit calibration, can be seen in Figure 17. Table 7 summarizes the results before and after calibration.

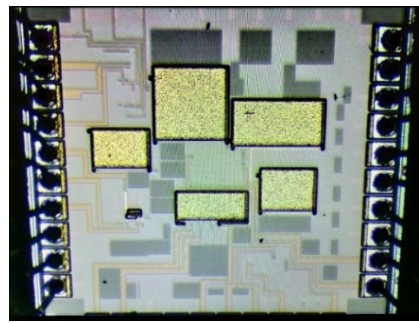

**Figure 15.** Chip fabricated in 130 nm BiCMOS technology.

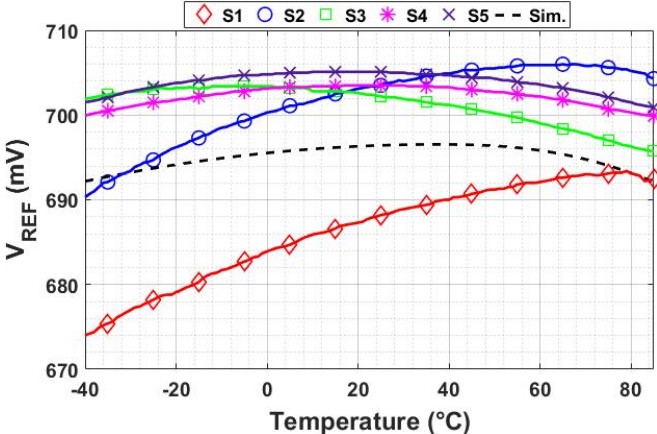

**Figure 16.** Measured $V_{REF}$ as a function of temperature for five chip samples (untrimmed).

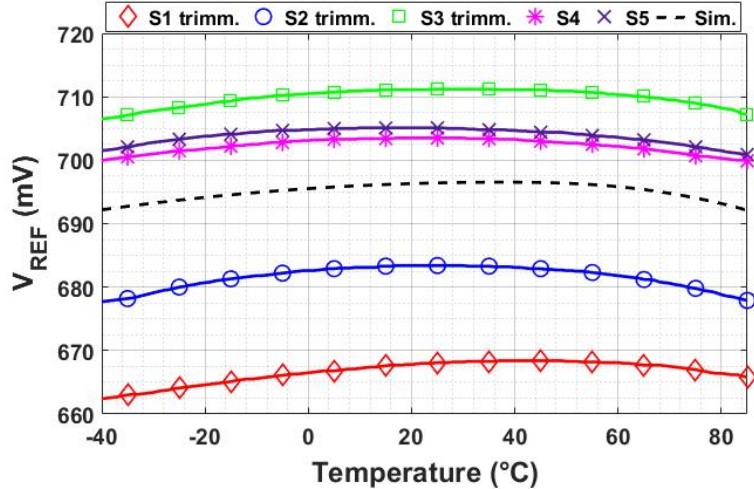

**Figure 17.** Measured $V_{REF}$ as a function of temperature for five chip samples (trimmed).

**Table 7.** Measured $V_{REF}$ as a function of temperature.

|  | $V_{REF}$ (mV) @ 27°C | TC (ppm/°C) | Trimmed $V_{REF}$ (mV) @ 27°C | Trimmed TC (ppm/°C) |
|---|---|---|---|---|
| Simulated | 696.4 | 50.8 | - | - |
| S1 | 688.4 | 225.4 | 668.1 | 71.8 |
| S2 | 703.8 | 178.4 | 683.4 | 66.7 |
| S3 | 702.1 | 87.7 | 711.2 | 52.8 |
| S4 | 703.5 | 40.9 | - | - |
| S5 | 705 | 47.6 | - | - |

Table 7 shows the efficiency of the trimming circuit. The mean TC before the calibration was 116 ppm/°C, while the corrected one was 56 ppm/°C with an improvement of 48%.

Figure 18a shows the measured $I_{REF}$ as a function of temperature, respectively. There is a variation of more than 20% in the value of $I_{REF}$ for S1. Such large variation was also predicted by the Monte Carlo analysis, and this results in a degraded temperature performance of $V_{REF}$, as seen in Figure 16. The choice of the bias current generator circuit topology was motivated by the absence of resistors and the small area occupation. If a precise bias current generator was used, the inclusion of the calibration circuit may not be needed.

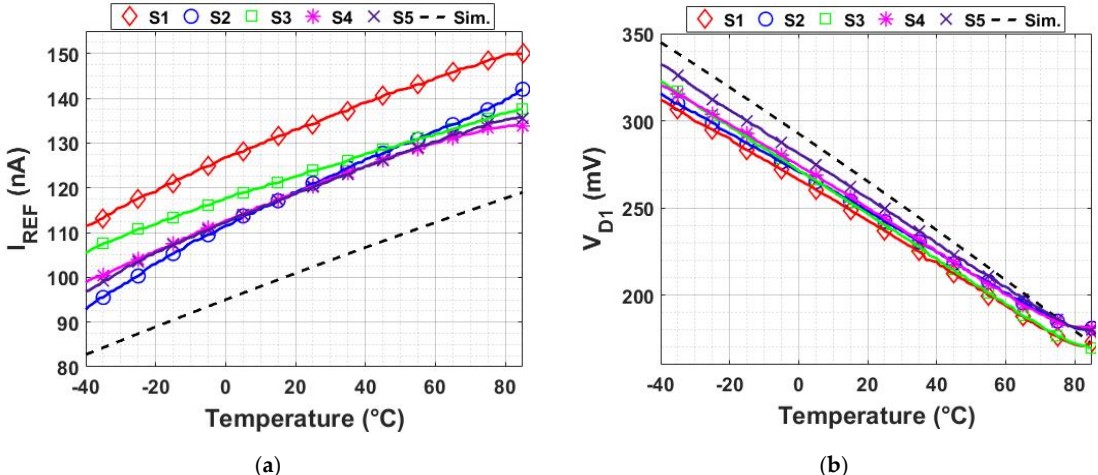

**Figure 18.** (**a**) The measured $I_{REF}$ and (**b**) the Schottky diode as a function of temperature.

Figure 18b shows the temperature behavior of the Schottky diode. As can be seen, the voltage has CTAT behavior and agrees with the simulation results.

Figure 19a,b shows the temperature dependence of the output voltage generated by the SCCTs. These results agree with the discussion of Section 2. In the proposed reference, the upper transistor of the SCCTs are always in saturation mode in order to guarantee the small dependence of the bias current. As presented in Section 2.3, a current level $I_{BIAS} > 10$ µA is required to have a linear PTAT relation between current and temperature. Since the supply current specification of the voltage reference was defined to be less than 1 µA, a different design approach was used. The linear PTAT relation was achieved using a large ratio of ($S_{UP}*I_{DS\_DOWN}/S_{DOWN}*I_{DS\_UP}$) according to Equation (6). Note that, in Table 3, that N5 has 18 unitary devices in parallel, while the N6 transistor as 24 unitary devices in series. This results in a $S_{UP}/{S_{DOWN}}$ ratio of 432, and $I_{DS\_DOWN} >> I_{DS\_UP}$.

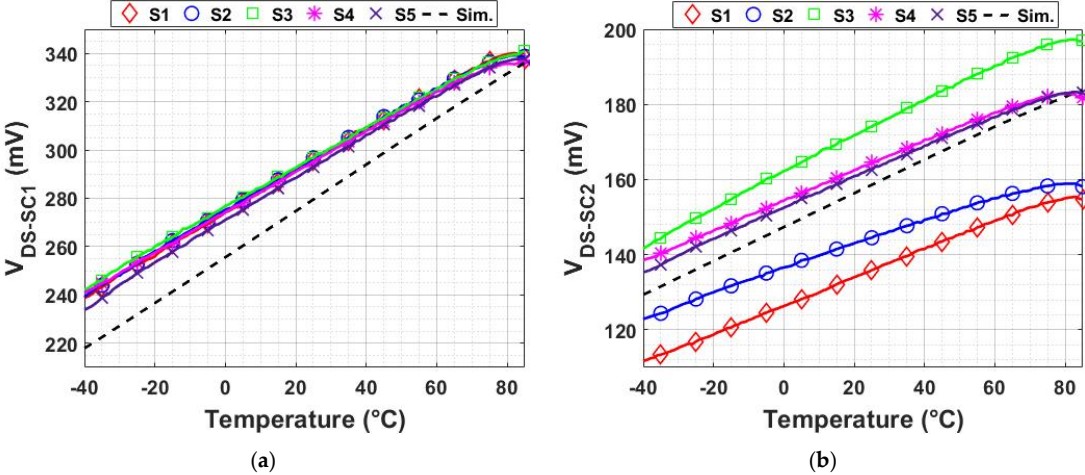

**Figure 19.** The measured output voltage of (**a**) SCCT1 and (**b**) SCCT2 as a function of temperature.

Figure 20 shows the V$_{REF}$ and $I_{REF}$ variation as a function of V$_{DD}$ variation for 26 samples in a range from 0 to 2.5 V, respectively. The simulated line regulation for V$_{REF}$ is 0.72 mV/V, while the mean value of the measured line regulation is 2.3 mV/V with a σ of 0.6 mV. For $I_{REF}$, the simulated line regulation is 0.24 nA/V, and the mean value of measured results is 0.9 nA/V, with a standard deviation of 0.7 nA.

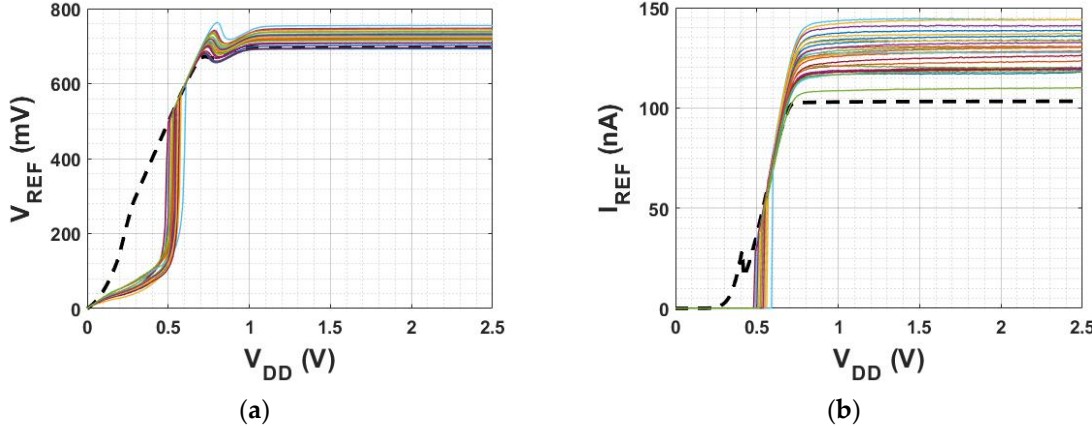

(a)                                                   (b)

**Figure 20.** Supply voltage variation of (**a**) V$_{REF}$ and (**b**) I$_{REF}$.

The supply voltage sweep shows a minimum V$_{DD}$ of operation close to 1.1 for V$_{REF}$ and 0.7 V for the bias current generator. The simulated power supply rejection ratio (PSRR) at DC is −55 dB and −90 dB for the supply voltages of 1.1 V and 2.5 V, respectively. The current supply and the power consumption (@ 1.1 V) of the proposed circuit are 680 nA and 748 nW, respectively. Finally, Table 8 summarizes the main measured results discussed so far.

**Table 8.** Summary of results.

| V$_{REF}$ @ 27°C | | TC_V$_{REF}$ | | Temp Range (°C) |
|---|---|---|---|---|
| Mean (mV) | σ (mV) | Mean (ppm/°C) | σ (ppm/°C) | |
| 720.4 | 16.6 | 56 | 13 | 125 |
| I$_{REF}$ @ 27°C | | TC_I$_{REF}$ | | V$_{DD}$ Range (V) |
| Mean (nA) | σ (nA) | Mean (ppm/°C) | σ (ppm/°C) | |
| 126.8 | 9.11 | 2500 | 452 | 1.1–2.5 |
| Line reg. (V-$_{REF}$) | | Line reg. (I-$_{REF}$) | | I$_{SUPPLY}$ (nA) |
| Mean (mV/V) | σ (mV/V) | Mean (nA/V) | σ (nA/V) | |
| 2.3 | 0.6 | 0.9 | 0.7 | 630 |

*Comparison to Other Works*

A comparison of this work with related works is presented in Table 9. The state-of-the-art voltage reference circuit topologies were chosen. As can be seen, considering the occupation of the silicon area, this work is, at least, 3.5 times smaller than the other works. Regarding the V$_{DD}$ current consumption, this work consumes at least 4.5 times less than the others. The designed circuit can achieve lower power consumption, without a large increase in the silicon area, because it does not employ integrated resistors. Moreover, the proposed topology is able to operate at 1.1 V of power supply.

**Table 9.** Comparison with state-of-the-art works.

|  | This Work | [14] | [15] | [16] | [17] | [18] |
|---|---|---|---|---|---|---|
| $V_{REF}$ (V) | 0.72 | 2.56 | 1.14 | 0.596 | 0.5 | 1 |
| Supply voltage (V) | 1.1 to 2.5 | 4.5 to 5.5 | 2 to 5 | 1.3 | 1.2 to 1.8 | 1.375 |
| Temp. range (°C) | −40 to 85 | −40 to 100 | −40 to 125 | −10 to 120 | 0 to 100 | −45 to 125 |
| Current consumption (μA) | 0.63 | 6.8 | 33 | 2.7 | 5.1 | 689 |
| Area (mm$^2$) | 0.011 | 0.075 | 0.0396 | 0.8 | 0.073 | 0.078 |
| TC (ppm/°C) | 56 | 2.6 | 1.01 | 30.95 | 22 | 6 |
| CMOS Process (nm) | 130 | 180 | 350 | 180 | 180 | 7 |
| Line regulation (mV/V) | 2.3 | 0.02 | 2 | - | 1.4 | 1 |
| Year | 2019 | 2019 | 2019 | 2019 | 2019 | 2019 |

Regarding the TC, all works show a superior performance. The proposed circuit topology may be a right circuit candidate for applications that do not need high precision but require a moderate/small silicon area and power consumption.

## 5. Conclusions

This work studied the temperature behavior of SCCT output voltage. The measurement data shows that this type of device can provide an output voltage with PTAT behavior. Moreover, with the proper sizing and biasing, the output voltage can also be set to be reasonably temperature-compensated. Moderate temperature compensation is obtained by biasing the upper transistor in triode mode, which results in an undesired dependency of the SCCT voltage on the bias current. If a precise current reference is available in the chip, a moderate temperature-compensated voltage can be easily obtained using SCCTs with a low silicon area and a low power consumption.

Driven by the obtained testing results of SCCTs, a new resistorless voltage reference circuit topology using a Schottky diode is also prototyped in this paper. The proper operation of the designed circuit was checked in 26 chip samples. The average TC is 56 ppm/°C in a temperature range from −40 to 85 °C. The trimming circuit was effective in improving the TC of $V_{REF}$.

**Author Contributions:** Conceptualization, design, analysis: T.M.B., D.M.C., and R.L.M.; measurement: T.M.B.; writing—original draft preparation: T.M.B. and D.M.C.; writing—review and editing, D.M.C., R.L.M., and K.E.S.; technical discussion: K.E.-S. and R.L.M.

**Funding:** This study was financed in part by the Coordenação de Aperfeiçoamento de Pessoal de Nível Superior - Brasil (CAPES) - Finance Code 001. It was also financed in part by the FAPEMIG (process - APQ-01758-18), and in part by the Universidade Federal de Minas Gerais (Edital PRPq - 03/2019). Moreover, there was a MOSIS chip support as well.

**Acknowledgments:** The authors are grateful to FAPEMIG, CAPES, and CNPq for financial support, MOSIS for the chip fabrication, and CTI Renato Archer for chip packaging. We also thank PPG-E (UNIFEI) and PPGEE (UFMG).

**Conflicts of Interest:** The authors declare no conflict of interest.

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
