# Peer review of "CMOS Voltage Reference Using a Self-Cascode Composite Transistor and a Schottky Diode"

_electronics, doi:10.3390/electronics8111271_

Round 1

Reviewer 1 Report

The main criticism of the work is the total absence of comparison between the results provided and those presented in similar works by other authors (among them, some mentioned in the bibliography).

In order to be able to evaluate the actual contributions of the research presented in this work, it is essential to compare the results obtained with those presented in other works with very similar objectives and technologies.

For example, in [1] the authors present results from a reference that also uses Self-Cascode Composite Transistors, obtaining better and much better results for TC and line regulation, respectively.

Some aspects that must be corrected, clarified or provided:

Definition of TC as shown in (9) is only useful if Vout is linear with T. As defined, it is misleading. For instance, and referring to Fig 4(a), if Tmax = 75ºC and Tmin = -40ºC, then TC ≈ 0 !! (since Vout_max=Vout_min) What it means for authors  "more PTAT" (as used in line 114, 120, etc)? A large range of temperatures where the voltage has a positive TC? Caption in Fig 4: SCCT16 must be SCCT8 Temperature behavior of Self-cascode Composite transistor depends on Z and K, a indicated in (8). What are the values of Z and K for test circuit shown in Fig 3? That's it, what are the aspect ratios of transistors P1 to P9? Figures 4 to 6 shows results for 3 values of IBIAS, but they lose significance if parameters Z and K are not known for SCCT1 to SCCT8. Reference [10] (line 191) is missing. What does m stands for in equation (15)? Y-axis range in Fig 16 must be reduced (for instance to 670mV - 710mV) to fully show temperature variation of VREF The rectangle with the labels for each experimental data curve must be take out of the graph area (Fig16 to 19). In some cases, it literally covers up the results that are to be shown. If the results in Figure 19 are compared with those in Figures 4 to 6, it appears that IBIAS should be clearly greater than 1uA, but table 8 indicates that ISUPPLY is 0.63uA. Isn't there an inconsistency? What is the power consumption of the proposed voltage reference? This is a paramount specification in this type of low voltage designs. The bibliography could be more extensive, to cover works on voltage references that have similar objectives in terms of technology and specifications to the one of the presented work.

Reviewer 2 Report

Authors used CMOS voltage reference using self-cascode and schottky diode structures. As far as I know, there are some similar approaches. However, authors did not address that even in the introduction sections. Therefore, authors need to provide more references in the introduction sections. Since there are too many CMOS voltage reference papers, authors need to provide more references. Proposed idea is new and measured performance is good. For my opinion, authors can publish the paper to address the following comments. However, there are extensive writing needed. Therefore, manuscript is major revision before acceptance since it takes time to be revised. 

1.Authors must mention which reference circuits have advantages and disadvantages. In the introduction section, there are too small numbers of the references. This is journal paper so authors need to provide literature reviews for that.

2.English grammar need to be carefully checked since there are too many English grammar errors. Therefore, authors need to use professional services or native colleague faculties. 

3.Reference format is not correct. Please see MDPI reference format.

4. Authors need to mention why L = 3um for 0.13 um process. Is this for temperature variance ? 

5. Why chose Wp = 60 um and Wn= 3um and Wp = 42 um and Wn= 16um. Authors need to provide clear description even though authors showed several simulation results.

6. Authors need to mention which kinds of ESD circuit for this circuit.

7.In Figure 16, why S1 has more changes compared to others.

8. In Figure 18 (a), S1 has some changes in Iref but has small changes in Vd1. Is there any reason ?

9.  Authors mentioned MOSIS fabrication. I guess there is MOSIS chip support. Therefore, it is better to mention that in Funding section if possible. Otherwise, it is fine.

10. In the conclusion section, authors need to mention important and critical results to support the proposed circuit.

Round 2

Reviewer 1 Report

The suggestions, concerns and corrections proposed by the reviewer have been satisfactorily taken into account.
My opinion is that, in its current form, the paper can be accepted without requiring further revisions.

Reviewer 2 Report

Authors well defended my questions so the paper can be fully accepted without any further revision.